# 12 weeks of strength training improves fluid cognition in older adults: A nonrandomized pilot trial

Timothy R. Macaulay[1]*, Judy Pa[2], Jason J. Kutch[1], Christianne J. Lane[3], Dominique Duncan[2], Lirong Yan[2], E. Todd Schroeder[1]

1 Division of Biokinesiology and Physical Therapy, Ostrow School of Dentistry, University of Southern California, Los Angeles, California, United States of America, 2 Mark and Mary Stevens Neuroimaging and Informatics Institute, Keck School of Medicine, University of Southern California, Los Angeles, California, United States of America, 3 Department of Preventive Medicine, Keck School of Medicine, University of Southern California, Los Angeles, California, United States of America

* tmacaula@usc.edu

**Data Availability Statement:** All relevant data are within the paper and its Supporting information files.

**Funding:** This study was supported by the Southern California Clinical and Translational

## Abstract

### Objectives

Resistance training (RT) is a promising strategy to slow or prevent fluid cognitive decline during aging. However, the effects of strength-specific RT programs have received little attention. The purpose of this single-group proof of concept clinical trial was to determine whether a 12-week strength training (ST) program could improve fluid cognition in healthy older adults 60 to 80 years of age, and to explore concomitant physiological and psychological changes.

### Methods

Twenty participants (69.1 ± 5.8 years, 14 women) completed this study with no drop-outs or severe adverse events. Baseline assessments were completed before an initial 12-week control period, then participants were re-tested at pre-intervention and after the 12-week ST intervention. The NIH Toolbox Cognition Battery and standard physical and psychological measures were administered at all three time points. During the 36 sessions of periodized ST (3 sessions per week), participants were supervised by an exercise specialist and challenged via autoregulatory load progression.

### Results

Test-retest reliability over the control period was good for fluid cognition and excellent for crystallized cognition. Fluid composite scores significantly increased from pre- to post-intervention (8.2 ± 6.1%, p < 0.01, d = 1.27), while crystallized composite scores did not (-0.5 ± 2.8%, p = 0.46, d = -0.34). Performance on individual fluid instruments, including executive function, attention, working memory, and processing speed, also significantly improved. Surprisingly, changes in fluid composite scores had small negative correlations with

Science Institute (SC-CTSI, grant number
UL1TR001855 to ETS and TRM). The SC-CTSI
(https://sc-ctsi.org/) did not play any role in the
study design, data collection and analysis, decision
to publish, or preparation of the manuscript.

**Competing interests:** The authors have declared
that no competing interests exist.

changes in muscular strength and sleep quality, but a small positive correlation with
changes in muscular power.

## Conclusions

Thus, improvements in fluid cognition can be safely achieved in older adults using a 12-
week high-intensity ST program, but further controlled studies are needed to confirm these
findings. Furthermore, the relationship with other widespread physiological and psychologi-
cal benefits remains unclear.

## Introduction

Older adults naturally experience declines in fluid cognition, the ability to process informa-
tion, solve novel problems, and encode new memories [1]. The rate of fluid decline is linked to
neurobiological integrity and neurological disorders that can interfere with independent living
[2]. However, higher levels of physical activity reduce the rate of cognitive decline and risk of
dementia [3], and exercise interventions have well-documented cognitive benefits [4]. Resis-
tance training (RT) is one such form of exercise. A recent meta-analysis reported positive
effects (d = 0.71) of RT interventions on cognition composite scores [5].

RT interventions can vary in their training parameters: exercise selection, intensity (load),
volume (sets and repetitions), frequency, and rest periods. Most of the previous RT studies for
cognition in healthy older adults involved low-intensity/high-volume circuits [6–8] or more
traditional moderate-intensity/moderate-volume hypertrophic weightlifting [9–14]. While
improvements in global fluid cognition and executive function appear to be partially mediated
by increases in muscular strength [15,16], it is unclear whether this relationship holds with a
strength-specific RT program. Knowing that a high-intensity periodized strength training (ST)
program can maximize gains in muscular strength [17,18], an investigation for cognitive-
enhancement is warranted.

Other pleiotropic benefits of RT are commonly observed in physiological and psychological
outcomes that are epidemiologically linked to cognition [19]. The most familiar effects are
increased physical capacity and protection against sarcopenia [20,21], but improvements are
also notable in measures of body composition, mood, sleep quality, and cardiovascular health
[22–24]. These factors may help us understand how cognitive enhancement arises. Even with
only small associations, there may be cumulative effects from each underlying factor.

The purpose of this proof-of-concept study was to estimate the effects of high-intensity ST
on fluid cognition in healthy older adults, and to explore potential relationships with concomi-
tant physical and psychological changes. The single arm pre-post design with initial control
period allowed us to address practicality, while maintaining a thorough proof-of concept
framework [25]. While various cognitive assessments have been used [5], we chose the NIH
Toolbox® Cognition Battery (NIHTB-CB) as a standardized set of comprehensive assessment
tools [26]. The NIHTB-CB has been normed and validated in participants ages 3–85, and
ensures that assessment methods and results can be used for comparisons across existing and
future studies [27]. We hypothesized that a 12-week ST program performed 3 days per week
could improve fluid cognition in healthy older adults. As an exploratory analysis, the wide-
spread benefits of ST were analyzed for potential associations with cognitive changes.

## Methods

### Participants and study design

Using a single arm pre-post design, the effects of a 12-week ST intervention on fluid cognition were compared to that of an initial control period. Thus, participants served as their own controls and completed all assessments at three time points: baseline, pre-intervention, and post-intervention. All control periods took place before the ST intervention to ensure that results were not confounded by detraining effects or long-term cognitive benefits [12]. In addition, a control period equal in duration to the intervention allowed direct within-subjects statistical comparisons. Participants were asked to not change their eating or exercise habits outside of the study and were encouraged to continue their normal activities.

All procedures of this single-group clinical trial were conducted at the University of Southern California (USC, Los Angeles, CA) Health Sciences Campus. This study was approved by the USC Health Sciences Review Board and registered with ClinicalTrials.gov (ID: NCT03982550). Clinical trial registration was performed retrospectively due to technical issues with the account. All future trials will be registered prospectively. A flow diagram of participant recruitment and testing is shown in Fig 1. Recruitment took place between July 2018 and May 2019. Strategies included participant referral and flyer distribution with, whenever possible, public announcement (e.g. at the end of a class at the local senior center). The USC Health Sciences Campus is approximately 5 miles from residential/business areas, which provide access to health clubs, country clubs, and senior centers. Only investigators (stating their association with the study and USC) discussed the study details with potential participants (including the time requirements, training protocols, and outcome measures) and inquired about possible interest.

Written informed consent was obtained from all potential participants assessed for eligibility. Primary considerations for inclusion/exclusion criteria were to ensure that participants were healthy, interested, and available to participate, without any contraindications to ST. In addition, participants were required to meet the following criteria: 60–80 years of age; living independently (without need of assistance); answer 'NO' to all questions on the Physical Activity Readiness Questionnaire (PAR-Q) or receive medical clearance from a physician; cognitively healthy (score 24 or higher on the Mini Mental State Examination); have no known history of neurological disease, cerebral infarct, or traumatic brain injury; have no known Type 1 or Type 2 Diabetes; and do not self-report engaging in heavy RT (i.e. "using a resistance heavy enough that you could not lift more than 15 times in a set") in the last 6 months.

To first confirm the effects on fluid cognition, we computed a preliminary sample size of 18 in an a priori power analysis. We used a meta-analysis effect size of RT on fluid cognition, d = 0.71 [5], alpha of 0.05, power of 0.80, and within-subjects pre-post design. We accounted for an anticipated attrition rate of 10%. All 20 participants (Table 1) that were included in the study completed the required number of training sessions (>90%) and all assessments at baseline, pre-intervention, and post-intervention. The last participant finished post-intervention testing in December 2019. There were no participant drop-outs or severe adverse events.

### Resistance training intervention

Participants performed a periodized and progressive ST program emphasizing development of total-body strength using machine-based exercises. All 36 training sessions (3 days per week) were performed in the USC Clinical Exercise Research Center, supervised by an exercise specialist, and lasted approximately one hour. Throughout the program, volume linearly decreased, and intensity linearly increased (Table 2). Mesocycle I (weeks 1–4) was designed for

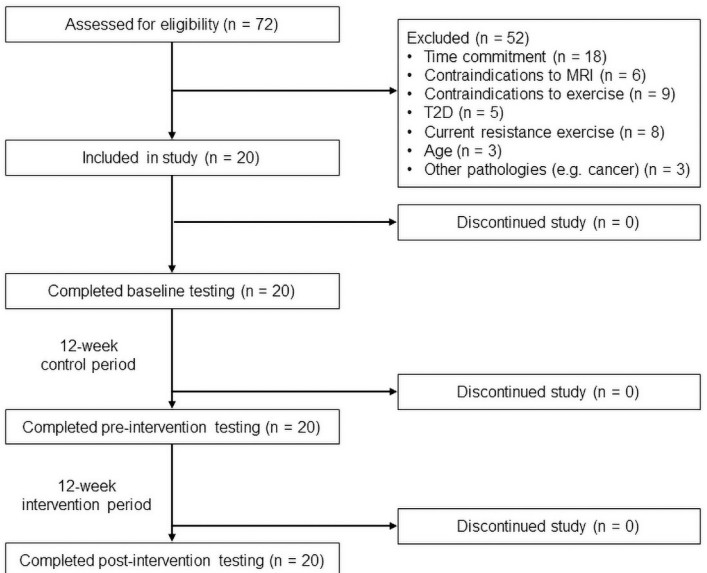

**Fig 1. Flow diagram of participant recruitment and testing.** 72 potential participants were assessed for eligibility. All 20 participants that were included in this study completed the required number of training sessions (>90%) and all assessments at baseline, pre-intervention, and post-intervention. There were no participant drop-outs or severe adverse events.

muscular hypertrophy to develop a base for more intense training in later phases. Mesocycles II (weeks 5–8) and III (weeks 9–12) were designed to promote strength gains. Training sessions alternated between lower body (leg press, leg extension, leg curl) focus and upper body (chest press, lat pulldown, seated row, and seated shoulder press) focus.

These parameters were chosen to maximize performance on 4–6 repetition maximum (RM) testing post-intervention, according to the principle of specificity [18]. Training loads

**Table 1. Participant characteristics at baseline (N = 20).**

| Variable | Actual |
|---|---|
| Female | 14 (70%) |
| Non-Hispanic: | 15 (75%) |
| • Asian | • 2 (10%) |
| • Black or African American | • 4 (20%) |
| • White | • 9 (45%) |
| Hispanic: | 5 (25%) |
| • White | • 1 (5%) |
| • More than one race | • 4 (20%) |
| Age (years) | 69.1 ± 5.8 (60.1,79.7) |
| Height (cm) | 166.7 ± 8.9 (154.4, 188.2) |
| Weight (kg) | 72.3 ± 10.1 (58.3, 96.5) |
| Mini Mental State Examination^ | 28.0 ± 1.6 (26, 30) |

N (%) or Mean ± SD (range).

^Score < 24 indicates possible mild cognitive impairment.

All participants were required to score ≥ 24 in order to be eligible for the study.

**Table 2. Periodization model of the high-intensity resistance training program.**

| | Macrocycle: 12 weeks | | | | | |
|---|---|---|---|---|---|---|
| | Mesocycle 1 | | Mesocycle II | | Mesocycle III | |
| Sets | 3 | 3 | 4 | 3 | 4 | 3 |
| Repetitions | 10 | 8 | 6 | 6 | 4 | 4 |
| Weeks | 1–2 | 3–4 | 5–6 | 7–8 | 9–10 | 11–12 |

This linear periodization model was used to maximize strength gains during the 12-week intervention.

were individually progressed in a safe and effective manner to elicit the greatest training-induced neuromuscular adaptations [28]. Because older adults respond to training stimuli at different rates, autoregulatory progression was used [29]. While the sets and repetitions were fixed for each week, supervised autoregulation allowed participants to progress loads at their own pace based on maximum performance on the last set of each exercise.

In the event a participant missed a training session during the 12-week intervention, she/he could complete missed sessions during a 2-week buffer period. A maximum of 6 sessions (3 per week) were allowed during this period. All post-intervention study procedures were conducted at least 48 hours after the last training session and within 2 weeks.

## Primary outcomes—Cognitive function

All cognitive assessments were administered by the same trained investigator (TRM) using the NIHTB-CB (Version 1.21) application on an iPad Pro 10.5-inch (Apple, Cupertino, CA). The NIHTB-CB has seven individual instruments—Flanker Test, Picture Sequence Memory Test, List Sorting Working Memory Test, Dimensional Change Card Sort Test, Pattern Comparison Test, Picture Vocabulary Test, and Oral Reading Recognition Test—that test the following cognitive subdomains: inhibitory control and attention, episodic memory, working memory, executive function, and processing speed for fluid cognition and vocabulary and reading recognition for crystallized cognition [26]. Each assessment lasted approximately 45 minutes.

In addition to scores for each individual instrument, the battery yields a Fluid Composite Score and a Crystallized Composite Score. The NIHTB-CB has good convergent validity with Gold Standard measures of Crystallized (r = 0.90) and Fluid (r = 0.78) Composite Scores and excellent test-retest reliability (r = 0.92 and 0.86 for Crystallized and Fluid Composite Scores, respectively), which is similar to that of the Gold Standard [27]. All scores are standardized to the population, with a normative mean of 100 and a standard deviation of 15.

## Secondary outcomes

**Body composition.**   Body composition was assessed in morning fasted conditions. Weight was measured on an InBody 770 (InBody, Seoul, South Korea). Percent body fat (PBF, %), lean body mass (LBM, g), and fat mass (FM, g) were measured by dual energy x-ray absorptiometry (DXA) on a Lunar iDXA (GE Healthcare, Waukesha, WI).

**Muscular strength.**   Participants performed a familiarization session approximately one week prior to baseline strength testing. They were instructed in proper technique on all machine-based resistance exercises to be used for testing and training. Muscular strength was assessed using 4-6RM tests. The load and number of completed repetitions were used to estimate 1RM values according to a standard equation. This calculated 1RM represents a valid assessment of muscular strength in older adults [30]. Single measures for lower body strength,

upper body strength, and total body strength were calculated using the cumulative 1RMs of their respective exercises.

**Physical function.**   Gait speed is an important marker of vitality that can predict the life expectancy of older adults [31]. Participants were instructed to walk on a marked 8-meter walkway at a normal, comfortable speed. Time taken to walk the central 4 meters of the course was recorded. The average of two measurements were used to calculate gait speed (m/sec).

The Timed Up and Go (TUG) test was used as a standard measure for mobility and fall risk [32]. Participants began seated in a chair with hands on the armrests and then were asked to rise, walk to a line on the floor 3 meters from the chair, turn around, and return to the same seated position as quickly and safe as possible. The average of three timed trials was recorded (sec).

Functional power was measured using the Margaria Stair Climb test for older adults [33]. We used a modified version in which participants were instructed to ascend a flight of 10 stairs one step at a time as quickly as possible without using the handrail. Timing began when one foot stepped on the 3rd stair and ended when that foot reached the 9th stair. Power (W) was then calculated using the stair height, body weight, and average time of three trials.

The Y-Balance test (Functional Movement Systems) was used as a dynamic assessment of single-leg standing balance abilities [34]. Participants stood with one foot on a platform from which 3 polyvinylchloride pipes extend in the anterior, posteromedial, and posterolateral directions. The participant was instructed to push a target with the reaching foot and return to a standing position without touching the ground. Maximal reach was recorded for each foot, in each of the 3 directions. The six reach distances were summed and used for analyses (cm). Four participants were not able to perform all six reach directions at baseline and/or pre-intervention, so their data were excluded from analyses.

**Questionnaires.**   Participants were asked not to include any activities of the present study in their questionnaire responses. The International Physical Activity Questionnaire (IPAQ) was used to assess average weekly physical activity outside of the study [35], determined in metabolic equivalents (MET*min/week). The Pittsburgh Sleep Quality Index (PSQI) was administered to evaluate overall sleep quality [36]. Scores are inversely related to sleep quality. A PSQI score of 5 or above is indicative of clinically poor sleep quality. The shortened Interpersonal Support Evaluation List (ISEL) was administered for perceptions of social support [37]. The ISEL has three dimensions of support: Appraisal support, Belonging support, and Tangible support. The three scores were averaged for an overall ISEL score. A higher ISEL score indicates greater social support, up to a maximum of 16. Finally, general health-related quality of life status was assessed using the Short Form-36 (SF-36). This survey yields an eight-scale profile of scores, including physical functioning, role limitations due to physical health, role limitations due to emotional problems, energy/fatigue, emotional well-being, social functioning, pain, and general health [38]. Higher scores indicate greater health status, up to a maximum of 100 for each scale.

## Statistical analyses

All raw data can be found in the S1 and S2 Datasets. The following intention to treat analyses were performed in SPSS Statistics V.25 (IBM, Chicago, IL). All data were considered continuous, examined for normality using Shapiro-Wilk tests, and described by either means and standard deviations or medians and interquartile range (if not normally distributed). For each outcome, all data from a participant were excluded if data from one visit were missing. Test-retest reliability over the control period (baseline and pre-intervention) was evaluated using two-way mixed effects intraclass correlation coefficients (ICCs) with absolute agreement.

Estimates less than 0.50, between 0.50 to 0.75, between 0.75 and 0.90, and greater than 0.90 were classified as poor, moderate, good, and excellent test-retest reliability, respectively [39].

Effect sizes were calculated for all outcomes to determine the magnitude of differences between control and intervention periods. For normally distributed data, an adapted Cohen's d effect size was calculated—mean changes from pre- to post-intervention were subtracted by the mean changes from baseline to pre-intervention, then divided by the average standard deviation of those changes [40]. For non-normally distributed data, matched-pairs rank-biserial effect size was calculated—the differences between control and intervention changes were ranked, then the sum of positive ranks was subtracted by the sum of negative ranks and divided by the total sum of ranks [41]. For both adapted Cohen's d and matched-pairs rank-biserial, effect sizes of 0.2, 0.5, 0.8, and 1.2 were classified as small, medium, large, and very large, respectively [42].

Only changes in cognition were tested for significance because fluid cognition was the only outcome powered for *a priori* hypothesis testing. Paired t-tests were used, comparing data from pre- and post-intervention, with statistical significance set at $P < 0.05$. The criteria for outlier exclusion were set at any z-score greater than 3 or less than -3. One outlier was noted for crystallized cognition. At post-intervention, this female participant had vocabulary and crystallized composite scores that were 4.3 and 4.1 standard deviations above the sample mean, respectively. Therefore, all crystallized cognition data for this participant were removed from further analyses.

Relationships between fluid cognition improvements and changes in other outcome variables were analyzed. Only raw changes from pre- to post-intervention were used. Spearman rank correlation coefficients were chosen to reduce effect variability with small samples [43]. Spearman's rho values of 0.1, 0.3, and greater than 0.5 were classified as small, medium, and large associations, respectively [44].

## Results

Tables 3 and 4 present the descriptive statistics for all outcome measures of the study: mean ± SD (or median ± IQR if data not normally distributed) at baseline, pre-intervention, and post-intervention, test-retest reliability of the 12-week control period, and effect sizes of the 12-week intervention period. Because participants were asked to maintain their normal eating and exercise habits outside of the study, no changes in outcome measures were expected after the control period. Reliability was moderate to good for fluid cognition measures (except the picture sequence memory test had poor reliability), good to excellent for crystallized cognition measures, excellent for body composition, good for blood pressure and heart rate, excellent for muscular strength and power, moderate for physical function, and good for PSQI and ISEL responses (Tables 3 and 4). The eight SF-36 scales ranged from poor to good reliability.

Responses to the IPAQ were not normally distributed (Shapiro-Wilk $P < 0.05$). However, there were no differences between baseline (median = 4012 MET*min/week, IQR = 5806 MET*min/week) and pre-intervention (median = 4318 MET*min/week, IQR = 5211 MET*min/week) responses, or between pre-intervention and post-intervention (median = 4439 MET*min/week, IQR = 4709 MET*min/week) responses. Therefore, participants appeared to have followed instructions to not change their exercise habits outside of the study.

### Cognitive outcomes

All participants completed the NIHTB-CB at baseline, pre-intervention, and post-intervention. Composite score data are presented in Fig 2A. Fluid composite scores significantly increased from pre- to post-intervention (+7.7 ± 5.5 Standard Units, p < 0.01). Note that this

**Table 3. Descriptive statistics for cognitive outcomes at baseline, pre-intervention, and post-intervention (N = 20).**

| Measure | Baseline (Week 0) | Pre-Intervention (Week 12) | Post-Intervention (Week 24) | ICC^ (3,1) | Effect Size^^ (d) |
|---|---|---|---|---|---|
| **Cognitive Outcomes:** | | | | | |
| NIHTB-CB Fluid Composite Score | 95.4 ± 8.2 | 95.9 ± 8.4 | 103.6 ± 8.7 | 0.77 | 1.27 |
| • Flanker | 95.8 ± 7.5 | 97.3 ± 6.2 | 101.4 ± 7.2 | 0.64 | 0.40 |
| • List Sorting | 96.5 ± 9.1 | 94.9 ± 10.2 | 102.9 ± 6.2 | 0.69 | 1.19 |
| • Dimensional Change Card Sort | 101.5 ± 7.5 | 102.7 ± 7.4 | 105.9 ± 8.3 | 0.81 | 0.35 |
| • Pattern Comparison | 95.8 ± 12.1 | 93.90 ± 12.8 | 104.5 ± 12.8 | 0.57 | 1.15 |
| • Picture Sequence | 98.2 ± 12.1 | 100.5 ± 10.8 | 102.5 ± 12.4 | 0.47 | -0.03 |
| NIHTB-CB Crystallized Composite Score[#] | 112.4 ± 7.2 | 112.8 ± 8.3 | 112.1 ± 7.0 | 0.91 | -0.34 |
| • Picture Vocabulary[#] | 113.5 ± 7.6 | 113.6 ± 9.0 | 112.6 ± 7.2 | 0.89 | -0.26 |
| • Oral Reading[#] | 110.4 ± 6.7 | 111.0 ± 6.9 | 110.8 ± 6.6 | 0.93 | -0.35 |

The control period 12-week test-retest reliability was important since this study was a single group clinical trial. No changes were observed from baseline to pre-intervention. The effects of the 12-week periodized RT intervention are evident via changes from pre- to post-intervention and calculated effect sizes. Mean ± SD.

^Two-way mixed intraclass correlation coefficient (ICC) with absolute agreement calculated using baseline and pre-intervention data.

^^Adapted Cohen's d effect size calculated by subtracting the mean changes from pre- to post-intervention by the mean changes from baseline to pre-intervention and dividing the average standard deviation of those changes.

[#]Data from one participant were considered outliers and therefore excluded from analyses (N = 19).

NIHTB-CB = NIH Toolbox Cognition Battery. All scores presented as uncorrected standard scores (population mean and SD = 100 ± 15).

mean 8.2% improvement is a very large effect size (d = 1.27) compared to the control period and is about one half of the population standard deviation (SD = 15 Standard Units) improvement. In contrast, there was no change in crystallized composite scores from pre- to post-intervention (-0.7 ± 3.3 Standard Units, p = 0.46). S1 Fig also presents the fluid composite score data at pre- and post-intervention for each participant, categorized by sex and age.

Data from each individual instrument are presented in Fig 2B. For fluid cognition, four of the five individual instrument scores significantly increased from pre- to post-intervention: Dimensional Change Card Sort Test (+4.1 ± 6.8 Standard Units, p = 0.04), Flanker Test (+3.2 ± 6.4 Standard Units, p = 0.02), List Sorting Test (+10.7 ± 10.2 Standard Units, p < 0.01), and Pattern Comparison Test (+2.0 ± 11.5 Standard Units, p < 0.01). The effect sizes were small for the Dimensional Change Card Sort Test and the Flanker Test, and large for the List Sorting Test and the Pattern Comparison Test (Table 3). The only fluid instrument that did not improve from pre- to post-intervention was the Picture Sequence Memory Test (+8.0 ± 8.5 Standard Units, p = 0.45). As expected for crystallized cognition, there were no changes from pre- to post-intervention in the Picture Vocabulary Test (-1.0 ± 4.5 Standard Units) or the Oral Reading Recognition Test (-0.2 ± 2.2 Standard Units).

## Secondary outcomes

Table 4 shows the descriptive statistics for physiological outcomes and questionnaire responses. Although these outcomes were not tested for statistical significance, all effect sizes were in the expected direction of improvement or were negligibly small. Fluid cognition improvements were explored for possible relationships with these outcomes. The SF-36 was excluded from these analyses because the median change for six of the eight domains was zero. Spearman correlations were computed between raw fluid composite score changes and changes in other outcome variables. Table 5 shows the degrees of correlation between the ranked data. Raw fluid cognition changes had small correlations with changes in fat mass, blood pressure, heart rate, total body strength, Margaria power, Y-balance, and PSQI score,

**Table 4. Descriptive statistics for physiological outcomes and questionnaire responses at baseline, pre-intervention, and post-intervention (N = 20).**

| Measure | Baseline (Week 0) | Pre-Intervention (Week 12) | Post-Intervention (Week 24) | ICC^ (3,1) | Effect Size^^ (d) |
|---|---|---|---|---|---|
| **Physiological Outcomes**: | | | | | |
| Body Weight (kg) | 72.3 ± 10.1 | 72.0 ± 9.5 | 72.2 ± 9.8 | 0.98 | -0.01 |
| Body Fat Percentage (%) | 38.8 ± 5.8 | 38.7 ± 5.9 | 37.8 ± 5.7 | 0.99 | 0.67 |
| Lean Body Mass (kg) | 42.8 ± 8.4 | 42.7 ± 7.9 | 43.5 ± 8.1 | 0.99 | 0.63 |
| Fat Mass (kg) | 26.8 ± 4.3 | 26.6 ± 4.5 | 26.2 ± 4.5 | 0.96 | 0.21 |
| Systolic Blood Pressure (mmHg) | 121.6 ± 18.9 | 121.1 ± 15.0 | 117.2 ± 14.9 | 0.79 | -0.04 |
| Diastolic Blood Pressure (mmHg) | 63.4 ± 10.4 | 64.2 ± 10.0 | 64.5 ± 10.9 | 0.77 | -0.14 |
| Heart Rate (bpm) | 65.7 ± 9.9 | 65.6 ± 11.1 | 64.5 ± 10.6 | 0.77 | -0.01 |
| Total Body Strength (kg) | 307.1 ± 141.3 | 299.5 ± 140.6 | 452.2 ± 162.6 | 0.99 | 3.86 |
| • Lower Body Strength (kg) | 207.6 ± 106.1 | 202.6 ± 106.1 | 319.9 ± 125.3 | 0.99 | 3.73 |
| • Upper Body Strength (kg) | 99.5 ± 37.4 | 96.8 ± 36.5 | 132.3 ± 40.9 | 0.98 | 3.37 |
| Margaria Power (W) | 362.2 ± 103.2 | 366.8 ± 101.6 | 417.2 ± 112.0 | 0.97 | 1.59 |
| Timed Up and Go (sec) | 5.9 ± 0.5 | 6.1 ± 0.6 | 5.4 ± 0.5 | 0.67 | -1.98 |
| Y-Balance## (cm) | 396.4 ± 38.7 | 407.7 ± 37.3 | 430.5 ± 38.3 | 0.59 | 0.55 |
| Habitual Gait Speed (m/sec) | 1.4 ± 0.2* | 1.4 ± 0.2* | 1.5 ± 0.2* | 0.68 | 0.01* |
| **Questionnaire Responses**: | | | | | |
| PSQI Score | 4.0 ± 2.8* | 5.0 ± 3.0* | 3.5 ± 1.75* | 0.84 | -0.39* |
| ISEL Score | 15.0 ± 2.0* | 14.3 ± 3.5* | 15.0 ± 2.3* | 0.81 | 0.69* |
| SF-36 Physical Functioning | 95.0 ± 13.8* | 92.5 20.0* | 95.0 10.0* | 0.53 | 0.47* |
| SF-36 Limitations Due to Physical Health | 100.0 ± 18.8* | 100.0 ± 25.0* | 100.0 ± 0.0* | 0.16 | 0.27* |
| SF-36 Limitations Due to Emotional Problems | 100.0 ± 24.9* | 100.0 ± 24.8* | 100.0 ± 0.0* | 0.10 | 0.30* |
| SF-36 Energy/Fatigue | 76.0 ± 9.8 | 76.8 ± 10.6 | 79.3 ± 10.4 | 0.56 | 0.19 |
| SF-36 Emotional Well-Being | 88.0 ± 8.0* | 88.0 ± 15.0* | 90.0 ± 8.0* | 0.77 | 0.26* |
| SF-36 Social Functioning | 100.0 ± 12.5* | 100.0 ± 0.0* | 100.0 ± 0.0* | 0.66 | -0.24* |
| SF-36 Pain | 90.0 ± 28.1* | 90.0 ± 30.0* | 90.0 ± 17.4* | 0.72 | 0.13* |
| SF-36 General Health | 77.8 ± 12.3 | 80.3 ± 12.5 | 85.3 ± 10.1 | 0.46 | 0.26 |

The control period 12-week test-retest reliability was important since this study was a single group clinical trial. No changes were observed from baseline to pre-intervention. The effects of the 12-week periodized RT intervention are evident via changes from pre- to post-intervention and calculated effect sizes. Mean ± SD.

^Two-way mixed intraclass correlation coefficient (ICC) with absolute agreement calculated using baseline and pre-intervention data.

^^Adapted Cohen's d effect size calculated by subtracting the mean changes from pre- to post-intervention by the mean changes from baseline to pre-intervention and dividing the average standard deviation of those changes.

*Median ± IQR and matched-pairs rank-biserial effect size calculated on the changes from pre- to post-intervention versus changes from baseline to pre-intervention.

##Four participants were not able to perform all six reach directions at baseline and/or pre-intervention, so their data were excluded from analyses (N = 16).

PSQI = Pittsburgh Sleep Quality Index. Lower scores indicate better sleep quality, scores ≥ 5 are classified as clinically poor sleep quality.

ISEL = International Support Evaluation List. Higher scores indicate greater social support, up to a maximum of 16.

SF-36 = Short Form-36. Higher scores indicate greater health-related quality of life, up to a maximum of 100.

and weak correlations with changes in body fat percentage, lean body mass, TUG time, and ISEL score. Spearman correlations were also computed between raw fluid composite score changes and pre-intervention values in other outcome variables, and are presented in S1 Table.

## Discussion

This study demonstrates that a 12-week ST intervention with cognitive outcomes is feasible and safe with healthy older adults. Although we recognize the limitations of the single-group design, this study was rigorously executed by qualified investigators using standard/well-

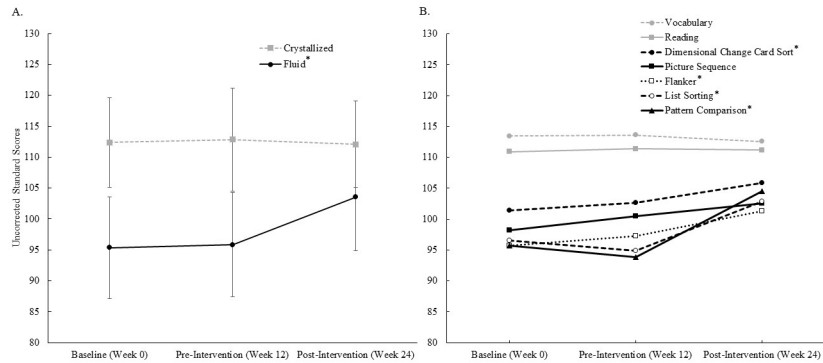

**Fig 2. NIH Toolbox Cognition Battery uncorrected standard scores assessed at baseline, pre-intervention and post-intervention.** A) Crystallized (N = 19) and fluid (N = 20) composite scores (Mean ± SD). B) Scores for each individual instrument (Means only, standard deviations were removed for simplicity). Crystallized composite scores and their corresponding instruments (Vocabulary and Reading) are shown in gray. Fluid composite scores and their corresponding instruments (Flanker, Dimensional Change Card Sort, List Sorting, Pattern Comparison, and Picture Sequence) are shown in black. * p < 0.05.

**Table 5. Spearman's rho correlations between pre- to post-intervention changes in fluid composite score and changes in other outcome variables of interest (N = 20).**

| Measure | Spearman's rho |
| --- | --- |
| Total Body Strength | -0.24 |
| • Lower Body | -0.25 |
| • Upper Body | -0.11 |
| Margaria Power | 0.21 |
| Body Fat Percentage | -0.07 |
| Lean Body Mass | 0.03 |
| Fat Mass | 0.17 |
| Systolic Blood Pressure | 0.11 |
| Diastolic Blood Pressure | -0.24 |
| Heart Rate | -0.17 |
| Timed Up and Go | 0.07 |
| Y-Balance[#] | 0.23 |
| Habitual Gait Speed | 0.01 |
| PSQI Score | 0.27 |
| ISEL Score | 0.01 |

These correlation analyses were used to explore factors that may influence fluid cognitive enhancement after RT. Raw pre- to post-intervention changes were used. Note that these correlations should be understood in the context of changes. For example, the small positive correlation with Margaria power suggest that greater increases in fluid composite score are associated with greater increases in Margaria power. However, the small negative correlation with total body strength suggest that greater increases in fluid composite score are associated with smaller increases in total body strength.

PSQI = Pittsburgh Sleep Quality Index. Lower scores indicate better sleep quality, scores ≥ 5 are classified as clinically poor sleep quality.

ISEL = International Support Evaluation List. Higher scores indicate greater social support, up to a maximum of 16.

[#]Four participants were not able to perform all six reach directions at baseline and/or pre-intervention, so their data were excluded from analyses (N = 16).

established techniques. Participants had a 100% adherence rate, and all operations were performed nominally (i.e., there were no severe adverse events or missed data collections). Findings should be interpreted with caution, but the data presented here can be used to support future studies with additional controls.

Significant improvements were observed in the NIHTB-CB fluid composite score and four of its five individual instruments: List Sorting Test, Pattern Comparison Test, Flanker Test, and Dimensional Change Card Sort Test. However, no changes were observed in the Picture Sequence Memory Test or in crystallized composite score and its instruments: Picture Vocabulary Test and Oral Reading Recognition Test. As expected with this type of exercise intervention, there were large effect on muscular strength and power, with notable positive effects on body composition and physical function. Although there were no changes in blood pressure or heart rate, positive mental health effects were observed in questionnaire responses, including improved sleep quality, perceptions of social support, and health-related quality of life. Together, these post-intervention improvements highlight the multidimensional impact that ST can have on overall health. Furthermore, the exploratory findings can contribute to our understanding of how fluid cognition is enhanced.

### Fluid cognition

In addition to cognition effect sizes, interpretations can be simplified in the context of aging. Population fluid composite scores tend to decline with age (r = -0.68) [27]. After adjusting for gender and education, age has a significant -0.0347 SD/year effect on fluid composite score (using population SD). In our study, the mean uncorrected fluid composite score improved by 7.7 standard units, or 0.51 SD, from pre- to post-intervention. Using the regression coefficient above, we calculate that the fluid cognition improvement observed in the current study is equivalent to 14.7 years of mean population fluid cognition decline. The supporting data (S1 Fig) showing differences in fluid composite score changes based on sex and age are interesting given previous reports of sex differences in age-related cognitive decline [45] and sex and age differences in exercise-related changes in cognition [46,47]. However, these comparisons should be interpreted with caution due to the small sample sizes.

Of the individual instrument mean score improvements, large effects were recorded for working memory and processing speed, and small effects were recorded for executive function and attention. The only subdomain that did not improve was episodic memory, but this measure also had the lowest reliability over the control period. This appears to both support and counter recent meta-analysis findings [5]. Landrigan and colleagues reported small to medium effects on executive functions (d = 0.39), but very small non-significant effects on working memory (d = 0.15). There are a couple of key explanations for these findings.

One reason could be small practice effects on the NIHTB-CB. In a test-retest reliability study with a sample representative of the population, the mean two-week (15.5 days) practice effect (d) for the NIHTB-CB fluid composite score was 0.42 [27]. In our study, the fluid composite practice effect was much smaller (z = 0.06), likely due to our older cohort (mean age = 69.1 years) and longer interval (12 weeks) between test-retest [48]. It is worth noting too that even less practice effects are expected for a third assessment [48].

Another reason could be the heterogeneity of cognitive measures used in the literature. Improvements in executive function have been commonly observed, with measures ranging from Flanker tests, Stroop tests, Trail Making tests, and Reaction Time tests [6,13,14]. Changes in memory with RT have been less consistent and may depend on the test administered; improvements have been observed in digit span tests [6,8,49], complex figure tests [49],

delayed word free recall [50], and verbal learning tests [12], but not in image recall tests [10] or auditory episodic memory tests [51].

There could also be discrepancies in the literature as to how measures are classified into fluid subdomains. Often measures will assess multiple subdomains. For example, working memory has dual service in executive control and episodic memory. Therefore, the NIHTB-CB treats working memory as a separate subdomain [26], with the List Sorting Working Memory Test designed on a paradigm emphasizing both holding and manipulation components. Other investigators, however, do not separate working memory from episodic memory. For example, in the recent meta-analysis [5], data from four studies that used the Rey Auditory Verbal Learning Test (RAVLT) were included in the working memory domain. But the RAVLT was used as a convergent validity measure ("Gold Standard") to validate the NIHTB-CB Picture Sequence Memory Test for episodic memory. Therefore, subdomain analyses require a more direct systematic approach before definite conclusions can be made about selective enhancement.

At least one other RT study has used the NIHTB-CB for fluid cognitive outcomes; however, they did not find any significant changes [7]. Instead, they found smaller effect sizes for fluid composite score (d = 0.22) and individual instrument scores (d = 0.03–0.51) compared to the current study, which may be partly due to differences in exercise prescription. Their circuit training protocol involved lower loads to optimize the production of muscular power for 12 repetitions per set, versus the high-intensity strength training model in the current study. They did, however, find improvements in executive function through improved inhibition (d = 1.49) and processing speed (d = 0.32) during ambulation [7].

## Exercise prescription

This study used a ST intervention, designed to maximize gains in muscular strength. These considerations might explain the large increase in global fluid cognition compared to that found in a recent meta-analysis (d = 1.27 versus d = 0.71, respectively), despite the duration (12 weeks) being shorter than the median duration (16 weeks, range 4–96 weeks) of the 24 studies in the meta-analysis [5]. Unfortunately, the effects of duration are currently unclear. Determining whether shorter interventions can produce similar results, or whether a plateau effect is present with longer durations can significantly affect exercise prescriptions.

The influence of other training parameters is also unclear. Most of the previous RT studies for cognition in healthy older adults involved low-intensity/high-volume circuits [6–8] or more traditional moderate-intensity/moderate-volume hypertrophic weightlifting [9–14]. Only one study [51] used a linear periodization model similar to what was used in the current study. However, the current study cycled down to lower reps (4 reps/set versus 6 reps/set), with correspondingly higher loads (90% of 1RM versus 85% of 1RM), that resulted in larger fluid effects (d = 1.27 versus d = 0.28). However, direct group comparisons are needed because previous studies reported no cognitive effects of higher intensity or frequency [9,14].

## Secondary and exploratory analyses

The effect on muscular strength was expected given the ST program. However, the negative correlations between changes in fluid composite score and changes in strength were unexpected. Previous studies have found the opposite—positive associations between changes in muscular strength and changes in fluid cognition [15]. Perhaps these differences are due to assessment techniques. While Mavros and colleagues assessed strength via pneumatic resistance machines for 1RMs, we used pulley- and plate-loaded resistance exercise machines for 4-6RMs. These machines have initial resistances that are not calculated as part of the load (e.g.

the sled on the leg press). Another difference in our study was the magnitude of strength gains. The effect size we observed was much larger than that by Mavros and colleagues (d = 3.86 versus d = 0.84, respectively). Therefore, there may be an unknown factor or interaction effect. Despite this unexpectedly negative correlation with strength, the positive correlation between changes in fluid composite score and changes in muscular power is consistent with the epidemiological link between physical and cognitive functions [19].

Habitual and acute sleep patterns play a vital role in long-term cognitive abilities and daily cognitive performance, respectively [52]. Age-related worsening of habitual sleep patterns is apparent with aging, and is associated with increased cognitive decline and greater risk of cognitive impairment [52]. But RT can improve sleep quality simultaneously with cognition [23], with greater effects observed when higher loads (i.e. > 70% 1RM) and/or higher frequencies (i.e. 3 days per week) are used [53]. Participants in our study had a median sleep quality score at pre-intervention (PSQI score = 5.0) that is considered to be the cutoff for clinically poor sleep quality [36]. But since our ST intervention involved high frequency and intensity (relative to Kovacevic and colleagues) it is no surprise that sleep quality was improved at post-intervention. A small correlation was observed between changes in fluid composite score and changes in sleep quality. The positive correlation indicates that smaller decreases in PSQI score (less improved sleep) were associated with larger increases in fluid composite score. This novel finding is small but interesting given the improvement in sleep quality from pre- to post-intervention. Future studies can refine our understanding of this relationship by designing combined interventions involving both RT and manipulated sleep patterns.

Last, we observed positive effects on other mental health measures. Improved perceptions of social support may serve as a buffer against the effects of stress [37]. And widespread improvements in certain scales of health-related quality of life (self-reported physical functioning, role limitations due to physical health, role limitations due to emotional problems, energy/fatigue, emotional well-being, pain, and general health) may represent an overall mood enhancement of RT [22]. However, changes were generally not correlated with cognition.

## Limitations

One limitation of this study was the use of a single group. Because this was a proof-of-concept study, the design was determined based on feasibility and study funding. Having a single-arm nonrandomized design allowed us to address practicality while maintaining a thorough proof-of-concept framework [25]. Furthermore, allowing all participants to receive the ST intervention may have contributed to the 100% adherence rate. However, blinding was not feasible with this design. Randomly crossing over the control and intervention periods was opposed so that results were not confounded by detraining effects or long-term cognitive benefits. While we were unable to conduct one-year follow up testing due to the COVID-19 pandemic [54], a previous study demonstrated that RT benefits in executive function and muscular power were still present at one-year follow up compared to the control group [12]. We expected similar benefits in our participants. Longer-term follow-up is needed to determine whether these interventions in healthy older adults can help prevent dementia. In addition, a separate active control group is needed in future studies to accurately estimate the effects of training and establish causal relationships.

## Conclusion

In conclusion, we have demonstrated the feasibility of conducting a ST intervention in older adults with promising analyses of cognitive outcomes. It appears that a high-intensity periodized ST program can improve fluid cognition in 12 weeks. Further controlled studies are

needed to confirm these findings. These relatively rapid improvements occurred simultaneously with well-characterized effects on muscular strength, physical function, body composition, sleep, and quality of life. Our exploratory analyses provide promising preliminary data for future studies to investigate the relationships between these changes, which may help us understand the shared mechanisms of benefit.

## Supporting information

**S1 Checklist. TREND statement checklist.**
(PDF)

**S1 Fig. Individual changes in fluid composite score from pre- to post-intervention based on A) sex and B) age.** A) Fluid composite scores increased from pre- to post-intervention in both females (+7.4 ± 6.2 Standard Units) and males (+8.5 ± 3.1 Standard Units). B) Using the median age at baseline (68.3 years), we compared the 10 participants below the median age (64.2 ± 2.5 years) and the 10 participants above the median age (74.0 ± 3.7 years). Fluid composite scores increased from pre- to post-intervention in both the participants below the median age (+8.7 ± 5.8 Standard Units) and the participants above the median age (+6.7 ± 4.9 Standard Units).
(TIF)

**S1 Table. Spearman's rho correlations between pre- to post-intervention changes in fluid composite score and pre-intervention values of outcome variables of interest (N = 20).**
(DOCX)

**S1 Dataset. Raw data for cognition composite scores, physical measures, and psychological measures.**
(XLSX)

**S2 Dataset. Raw uncorrected data for each cognitive instrument at each time point (baseline, pre-intervention, and post-intervention).**
(XLSX)

**S1 File.**
(DOCX)

## Acknowledgments

We thank our participants and student volunteers in the CERC for their dedication and hard work throughout the study.

## Author Contributions

**Conceptualization:** Timothy R. Macaulay, Judy Pa, Jason J. Kutch, Christianne J. Lane, Dominique Duncan, Lirong Yan, E. Todd Schroeder.

**Data curation:** Timothy R. Macaulay.

**Formal analysis:** Timothy R. Macaulay.

**Funding acquisition:** Timothy R. Macaulay, Judy Pa, Jason J. Kutch, Christianne J. Lane, Dominique Duncan, Lirong Yan, E. Todd Schroeder.

**Investigation:** Timothy R. Macaulay, E. Todd Schroeder.

**Methodology:** Timothy R. Macaulay, Judy Pa, Jason J. Kutch, Christianne J. Lane, Dominique Duncan, Lirong Yan, E. Todd Schroeder.

**Project administration:** Timothy R. Macaulay, E. Todd Schroeder.

**Resources:** Judy Pa, E. Todd Schroeder.

**Supervision:** E. Todd Schroeder.

**Visualization:** Timothy R. Macaulay.

**Writing – original draft:** Timothy R. Macaulay.

**Writing – review & editing:** Timothy R. Macaulay, Judy Pa, Jason J. Kutch, Christianne J. Lane, Dominique Duncan, Lirong Yan, E. Todd Schroeder.

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
