## [Decision Letter · Decision Letter 0]

10 May 2021

PONE-D-20-21856

12 weeks of strength training improves fluid cognition in older adults: a nonrandomized pilot trial

PLOS ONE

Dear Dr. Macaulay,

Thank you for submitting your manuscript to PLOS ONE. After careful consideration, we feel that it has merit but does not fully meet PLOS ONE’s publication criteria as it currently stands. Therefore, we invite you to submit a revised version of the manuscript that addresses the points raised during the review process.

Please address the methodological and formal comments related to the statistical aspects of the paper, as well as the other minoe points raised by the other reviewer.

We look forward to receiving your revised manuscript.

Kind regards,

Andrea Martinuzzi

Academic Editor

PLOS ONE

Journal Requirements:

Reviewers' comments:

Reviewer's Responses to Questions

**Comments to the Author**

1. Is the manuscript technically sound, and do the data support the conclusions?

Reviewer #1: Partly

Reviewer #2: Yes

Reviewer #3: Yes

2. Has the statistical analysis been performed appropriately and rigorously? 

Reviewer #1: I Don't Know

Reviewer #2: Yes

Reviewer #3: Yes

3. Have the authors made all data underlying the findings in their manuscript fully available?

Reviewer #1: Yes

Reviewer #2: Yes

Reviewer #3: Yes

4. Is the manuscript presented in an intelligible fashion and written in standard English?

Reviewer #1: Yes

Reviewer #2: Yes

Reviewer #3: Yes

5. Review Comments to the Author

Reviewer #1: Important note: This review pertains only to ‘statistical aspects’ of the study and so ‘clinical aspects’ [like medical importance, relevance of the study, ‘clinical significance and implication(s)’ of the whole study, etc.] are to be evaluated [should be assessed] separately/independently. Further please note that any ‘statistical review’ is generally done under the assumption that (such) study specific methodological [as well as execution] issues are perfectly taken care of by the investigator(s). This review is not an exception to that and so does not cover clinical aspects {however, seldom comments are made only if those issues are intimately / scientifically related & intermingle with ‘statistical aspects’ of the study}. Agreed that ‘statistical methods’ are used as just tools here, however, they are vital part of methodology [and so should be given due importance].

COMMENTS: Your ABSTRACT is well drafted but assay type. Please note that it is preferable [refer to item 1b of CONSORT checklist 2010: Structured summary of trial design, methods, results, and conclusions] to divide the ABSTRACT with small sections like ‘Objective(s)’, ‘Methods’, ‘Results’, ‘Conclusions’, etc. which is an accepted practice of most good/standard journals [including PLOS]. It will definitely be more informative then, I guess, whatever the article type may be.

Since it is a ‘proof of concept clinical trial’, sample size is not an issue. However, I am sure that the authors are aware of the well-known drawbacks of a single-arm design [a type of Quasi-experimental research], and it is often said that ‘alright to have ‘single-arm design’ (before-after study) when that is the only possibility’. It is very essential to keep those limitations in mind while interpreting results. Further, note that a classical/ideal clinical trial/study needs/requires a concurrently {but similarly} handled/treated appropriately selected/chosen control/comparison parallel group/arm.

Though measures/tools used are appropriate, most of them {may be even assessments tools used in this study} yield data that are in [at the most] ‘ordinal’ level of measurement [and not in ratio level of measurement for sure {as the score two times higher does not indicate presence of that parameter/phenomenon as double (for example, a Visual Analogue Scales VAS score or say ‘depression’ score)}]. Then application of suitable non-parametric test(s) is/are indicated/advisable [even if distribution may be ‘Gaussian’ (i.e. normal)].

When data are likely to be in ‘ordinal’ level of measurement, there is no sense in testing for ’normality’ (line 236: All data were considered continuous, examined for normality). Aany ’normality’ test [including Shapiro-Wilk test] cannot be powerful enough when sample size is this small. Moreover, I do not understand ‘why the differences between baseline and pre-intervention (lines 239-240 but later even differences from post-intervention) were evaluated using two-way mixed effects intraclass correlation coefficients (ICCs)’ instead of by direct comparison methods [like Wilcoxon’s Signed Ranked test – non-parametric equivalent/parallel of Paired t-test. {In my knowledge ‘differences’ are not evaluated using intraclass correlation coefficients (ICCs)} Though later (in lines 254-55) it is said that ‘Paired t-tests were used, comparing data from pre- and post-intervention’, however why ICCs’ are given in tables’? This is very confusing, in my opinion.

Titles & contents of Table 3 (Group means for cognitive outcomes at baseline, pre-intervention, and post-intervention) and Table 4 (Group means for physiological outcomes and questionnaire responses at baseline, pre-intervention, and post-intervention) are very confusing because they display mean and SD but in 4th and 5th column ICC and Effect Size (d) [where effect sizes are based on mean and the average standard deviation], are reported. They are in fact completely beyond my understanding [this may be very valid but definitely not according/adhering to current, known practice]. If it so, explain adequately.

Table 5 (Spearman’s rho correlations between pre- to post-intervention changes in fluid composite score and other outcome variables of interest) is similarly confusing. It is said in lines 263-4 that ‘Spearman’s rho values of 0.1, 0.3, and greater than 0.5 were classified as small, medium, and large associations, respectively’ and quoted reference [44] but I think this classification is for ‘effect size’ and not directly applicable for absolute values of ‘Spearman’s rho’. Is not that so? {please, again look at reference 44 to confirm}. Is what is said in lines 369-70 that “For the other outcome variables, pre-intervention values and raw pre- to post-intervention changes were used” correct? Readers should clearly know/understand regarding ‘What exactly is done?’. {Do you mean three items/values (Raw pre- to post-intervention changes in fluid composite score, pre-intervention values and raw pre- to post-intervention changes w.r.t. the other outcome variables) were/are used simultaneously?}

However, it should be noted that [despite ‘Limitations’ narrated in lines 496-509], this small study has many good points {use of appropriate tools, painstaking/good execution, etc., etc.} and has excellent ‘potential’ [only needs revision in light of above pointed out items], in my considered opinion.

Reviewer #2: Dear authors,

Congratulations for the work done. The paper is wonderfully written, and all analysis performed are perfectly convenient to answer the scientific questions raised. In addition, I find the research topic of a crucial interest for the proper development of the resistance training and health research field. Therefore, I proposed highlighting the manuscript at the PLOS ONE homepage.

Reviewer #3: In this manuscript, Macaulay et al. investigate whether strength training can improve fluid cognition in healthy older adults. It is well-written and does a great job of highlighting existing literature in the context of the current study. Their overall conclusions that a 12-week strength training program improves fluid cognition but not crystallised cognition are well-supported by their results and have far-reaching clinical implications. Overall, I found this to be an interesting study. Please find below some suggestions to improve the quality of the manuscript.

Major Comments:

- 70% of participants in this study are female. Sex differences have been reported in age-related cognitive decline (1) and in exercise-related changes in cognition (2). While this exploratory analysis may not be powered for explicit hypothesis testing related to sex differences, it would be interesting to see sex-specific results reported for this study. Authors could perform this by creating an additional (supplemental) figure replicating Figure 2 and demonstrating the trends for crystallised and fluid composites and individual tasks scores separately for males and females, and briefly discussing any observed differences in the discussion.

- Authors report that participants did not appear to change their exercise habits outside of the study (Page 16, lines 321-322). However, it would be interesting to know whether participants with a higher baseline exercise activity outside of the study had stronger improvements in their fluid cognition.

- Authors note that participants were allowed to progress loads at their own pace (Page 8, lines 156-157), but do not investigate whether this load progression is correlated with change in fluid cognition. Correlating the load progression with changes in fluid cognition would provide insight into a potential mechanism underlying the exercise-related improvements in fluid cognition. This may also provide more insight into the unexpected negative correlations observed between changes in muscular strength and changes in fluid cognition.

Minor Comments:

- In table 5, IPAQ is defined in the caption, but correlation for IPAQ itself doesn’t seem to be included in the table.

- In table 5, it might be helpful to clarify what the directionality of the correlations indicate in the caption.

- On Page 19, lines 400-402, authors say that fluid improvement observed is equivalent to 14.7 years of mean population fluid decline, but earlier in the paragraph, they state that yearly decline is -0.104 points/year. Based on the yearly decline, wouldn’t an increase of 1.53 points (0.104*14.7) be equivalent o 14.7 years of decline?

- The resolution of Figure 2 makes it difficult to read properly.

1. McCarrey, A. C., An, Y., Kitner-Triolo, M. H., Ferrucci, L., & Resnick, S. M. (2016). Sex differences in cognitive trajectories in clinically normal older adults. Psychology and aging, 31(2), 166.

2. Barha, C. K., Davis, J. C., Falck, R. S., Nagamatsu, L. S., & Liu-Ambrose, T. (2017). Sex differences in exercise efficacy to improve cognition: a systematic review and meta-analysis of randomized controlled trials in older humans. Frontiers in neuroendocrinology, 46, 71-85.

6. PLOS authors have the option to publish the peer review history of their article (what does this mean?). If published, this will include your full peer review and any attached files.

Reviewer #1: No

Reviewer #2: No

Reviewer #3: **Yes: **Elvisha Dhamala

---

## [Author Response · Author response to Decision Letter 0]

8 Jun 2021

Please see the Response to Reviewers file attached to this submission.

---

## [Decision Letter · Decision Letter 1]

9 Jul 2021

12 weeks of strength training improves fluid cognition in older adults: a nonrandomized pilot trial

PONE-D-20-21856R1

Dear Dr. Macaulay,

We’re pleased to inform you that your manuscript has been judged scientifically suitable for publication and will be formally accepted for publication once it meets all outstanding technical requirements.

Kind regards,

Andrea Martinuzzi

Academic Editor

PLOS ONE

Additional Editor Comments (optional):

Reviewers' comments:

Reviewer's Responses to Questions

**Comments to the Author**

1. If the authors have adequately addressed your comments raised in a previous round of review and you feel that this manuscript is now acceptable for publication, you may indicate that here to bypass the “Comments to the Author” section, enter your conflict of interest statement in the “Confidential to Editor” section, and submit your "Accept" recommendation.

Reviewer #1: All comments have been addressed

Reviewer #3: All comments have been addressed

2. Is the manuscript technically sound, and do the data support the conclusions?

Reviewer #1: (No Response)

Reviewer #3: Yes

3. Has the statistical analysis been performed appropriately and rigorously? 

Reviewer #1: (No Response)

Reviewer #3: Yes

4. Have the authors made all data underlying the findings in their manuscript fully available?

Reviewer #1: (No Response)

Reviewer #3: Yes

5. Is the manuscript presented in an intelligible fashion and written in standard English?

Reviewer #1: (No Response)

Reviewer #3: Yes

6. Review Comments to the Author

Reviewer #1: COMMENTS: All the comments made on earlier draft by me (and hopefully by other respected reviewers also) were/are addressed [though I am not fully satisfied by the answers/arguments]. and recommend acceptance (only) because [it is said that] the study was rigorously executed by qualified investigators using standard methods/techniques.

Reviewer #3: (No Response)

7. PLOS authors have the option to publish the peer review history of their article (what does this mean?). If published, this will include your full peer review and any attached files.

Reviewer #1: No

Reviewer #3: **Yes: **Elvisha Dhamala

---

## [Editor Report · Acceptance letter]

13 Jul 2021

PONE-D-20-21856R1 

12 weeks of strength training improves fluid cognition in older adults: a nonrandomized pilot trial 

Dear Dr. Macaulay:

I'm pleased to inform you that your manuscript has been deemed suitable for publication in PLOS ONE. Congratulations! Your manuscript is now with our production department. 

Kind regards, 

on behalf of

Dr. Andrea Martinuzzi 

Academic Editor

PLOS ONE